# Design and Verification of a Single-Channel Pump Model based on a Hybrid Optimization Technique

**Jin-Hyuk Kim [1,2,\*], Sang-Bum Ma [3], Sung Kim [1,2], Young-Seok Choi [1,2] and Kwang-Yong Kim [3]** 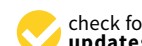

[1] Thermal & Fluid System R&D Group, Korea Institute of Industrial Technology 89 Yangdaegiro-gil, Ipjang-myeon, Seobuk-gu, Cheonan, Chungcheongnam-do 31056, Korea; ks2928@kitech.re.kr (S.K.); yschoi@kitech.re.kr (Y.-S.C.)

[2] Industrial Technology (Green Process and Energy System Engineering), Korea University of Science & Technology, 217 Gajeong-ro, Yuseong-gu, Daejeon 34113, Korea

[3] Department of Mechanical Engineering, Inha University, 100 Inha-ro, Michuhol-Gu, Incheon 22212, Korea; msb927@inha.edu (S.-B.M.); kykim@inha.ac.kr (K.-Y.K.)

\* Correspondence: jinhyuk@kitech.re.kr

**Abstract:** This paper handles a hybrid multiple optimization technique to concurrently enhance hydraulic efficiency and decrease unsteady radial forces resulting from fluid-induced vibration of a single-channel pump for wastewater treatment. A single-channel impeller and volute was optimized systematically by using a hybrid particle swarm optimization and genetic algorithm coupled with surrogate modeling. Steady and unsteady Reynolds-averaged Navier–Stokes analyses were conducted to optimize the internal flow path in the single-channel pump. Design variables for controlling the internal flow cross-sectional area of the single-channel impeller and volute in the single-channel pump were chosen to concurrently optimize objective functions with hydraulic efficiency and the unsteady radial forces resulting from impeller–volute interaction. The optimization results clearly showed that the arbitrary cluster optimum design considerably enhanced hydraulic efficiency and decreased the unsteady radial forces concurrently, compared to the reference design. Finally, the hydraulic performance of the optimized prototype model was verified experimentally. Then, it was proved that the proposed technique is a practical tool for designing a single-channel pump.

**Keywords:** single-channel impeller; computational fluid dynamics (CFD); unsteady RANS; optimization; hybrid PSO-GA; radial force

---

## 1. Introduction

Recently, the water industry has been receiving global attention as an industry with potential growth in the future. Current trends indicate that the wastewater treatment industry is a rapidly growing industry globally. Thus, the demand for special pumps for transferring sewage containing solid wastes is increasing rapidly. However, grinder and vortex pumps that are mostly used for wastewater treatment are inefficient and cannot transfer sewage with solid wastes because of low efficiency and the need for frequent maintenance, among other factors. To overcome these problems, many research studies have been conducted on treating sewage containing solid wastes [1–4]. One such study proposed a single-channel impeller with a single free annulus passage, which can smoothly transfer sewage containing solid wastes. Moreover, it has a high efficiency and wide flow range compared with those of grinder and vortex pumps. However, the single-channel impeller has an unsymmetrical flow passage, and there is a limit to stabilizing unstable vibrations with the generated radial forces due to unsteady features between the stationary volute and rotating impeller. Moreover, a single-channel pump impeller produces unstable radial force sources at a general frequency specified by the impeller rotating speed [4].

Researches on the interaction between the impeller and stator have been conducted over the past several years. Qin and Tsukamoto [5] analyzed the unsteady flow characteristics caused by impeller–diffuser interaction in a diffuser pump using a singularity method. The pressure fluctuation characteristics generated by rotor–stator interaction in a diffuser pump were investigated numerically by Shi and Tsukamoto [6]. As the result, it was confirmed that the flow characteristics generated by the impeller–diffuser interaction can be analyzed through an unsteady flow analysis. The unsteady flow characteristics between the radial impeller and diffuser in a radial pump were investigated by Feng et al. [7]. The flow between the impeller and diffuser was analyzed effectively by using the unsteady Reynold-averaged Navier–Stokes (RANS) analysis and laser Doppler velocimetry (LDV), and it was reported that there are two types of interaction effects between a rotor and stator. One was the downstream effect induced by the impeller, which has an unsteady flow characteristic because of the highly distorted flow field and the wake of the impeller. The other was the upstream effect induced by the stator, which induces unsteady pressure fluctuations. Experimental and computational numerical studies for investigating unsteady hydrodynamic forces on the diffuser pump impeller were performed by Zhang and Tsukamoto [8]. The hydrodynamic unsteady forces acting on the pump impeller were analyzed by varying the number of impeller blades. It was found that the hydrodynamic force was smaller when the blade number of impellers was equal to the diffuser vane number. However, pressure fluctuation was the largest compared with that in other cases. Benra [9] measured the hydrodynamic forces acting on the blade of a single-channel pump. Fluid-structure interaction (FSI) was used for numerical analysis of hydrodynamic forces, and the deflection of the blade was measured using proximity sensors. The vibration induced by the blade was transmitted to the volute and pipes. This phenomenon occurred periodically according to the rotation of the blade, and it was confirmed that the hydrodynamic force increases as the flow rate of the pump increases. Nishi et al. [10] proposed a single-channel pump design to increase the amount of solid waste passing through the impeller and to improve the pump performance. The results of the study showed that the pump efficiency exceeded 62%. Pei et al. [11] analyzed the flow characteristics of a single-blade centrifugal pump through numerical analysis and confirmed that unsteady flow characteristics are distributed according to the position of the blade. In addition, it has been reported that the stability of a single-channel pump can be improved by optimizing the flow characteristic or flow path between the blade and volute. These unsteady radial force sources were generally induced by the impeller–volute interaction, and consequentially, these sources adversely affected the entire hydraulic and mechanical performances (durability, life expectancy, etc.) of a single-channel pump. Therefore, an advanced design technique that can concurrently enhance hydraulic efficiency and decrease these unsteady sources is required.

To achieve multiple goals simultaneously, optimization methods using numerical analysis are widely used. Classical methods such as decomposition, Lagrangian techniques and cutting plane methods were previously applied to find optimal points. However, these classic optimization methods are ineffective for practical engineering problems. Hence, many studies were focused on exploratory optimization algorithms, such as a particle swarm optimization (PSO) [12] and genetic algorithm (GA) [13] to identify global optima, because the classic methods can be trapped in local optima according to the initially set value. These exploratory optimization methods combined with three-dimensional (3-D) RANS analysis have been generally used to find the best design solutions in the last decades—especially for turbomachinery designs [14,15]. Recently, a hybrid optimization algorithm was developed that combines the advantages of several optimization algorithms to improve computational efficiency. Robinson et al. [16] sought to combine GA and PSO and introduced an initial model of hybrid PSO-GA. Shi et al. [17] introduced an improved hybrid PSO-GA algorithm. The hybrid PSO-GA and GA were used to calculate benchmark test problems, and the performances of the algorithms were compared. The hybrid PSO-GA algorithm showed an improved performance with respect to the ability to identify optimal solutions and the speed of the convergence. Gandelli et al. [18] established another hybrid PSO-GA algorithm by applying a novel combinations strategy and tested

its performance using various benchmark test problems. The optimization using this hybrid PSO-GA algorithm led to consistent and effective results for the benchmark problems.

In fact, as shown in the above literatures, even though many computational optimization methodologies on various pump designs have been proposed, no systematic design works have been attempted for single-channel pumps. In the present work, the geometric shapes of the volute and impeller flow paths of a single-channel pump were systematically optimized to concurrently enhance hydraulic efficiency and decrease the unsteady radial forces by using hybrid PSO-GA based on a surrogate model. Here, for multi-objective optimization, the hydraulic efficiency and unsteady radial forces resulting from impeller–volute interaction were used as objective functions. These were then approximated using a radial-bias neural network (RBNN) model [19]. Furthermore, as one of the design-of-experiment (DOE) methods, Latin hypercube sampling (LHS) [20] was employed to efficiently choose the design points within the design space. This work is the follow-up study of the authors' previous works [21–23]. In the previous works, the authors verified that the design technique combined with the numerical analysis and optimization method is a practical tool for realizing the high-efficiency and low-fluid-induced vibration of the single-channel pump. Based on these results, the importance for objective functions was understood. Then, two objective functions were selected to design the high-efficiency and low-fluid-induced vibration single-channel pump. In addition to the hybrid PSO-GA, the advanced technique is used to improve the prediction accuracy and performance of the optimal design. The hybrid PSO-GA proved to be superior through some benchmark function tests, but rarely applied to the engineering problems [16–18]. Therefore, in this study, it was verified whether the hybrid PSO-GA technique could be practically used in the actual pump design step.

## 2. Numerical Methods

### 2.1. Single-Channel Pump Model

In this study, from the Stepanoff theory applied in the authors' preceding works [21–24], the initial geometry of the volute and impeller of the single-channel pump was designed, as shown in Figure 1. The Stepanoff theory is a method of controlling the cross-sectional area to minimize the flow losses caused by the flow rate differences. The Stepanoff theory is normally used in a volute design, but in this study it was used for designing both the volute and impeller because they each consist of a single flow path. Therefore, the initial area distributions of the volute and impeller are continuously increased as the theta angle between the volute and impeller increases from the inlet to outlet, as shown in Figure 2.

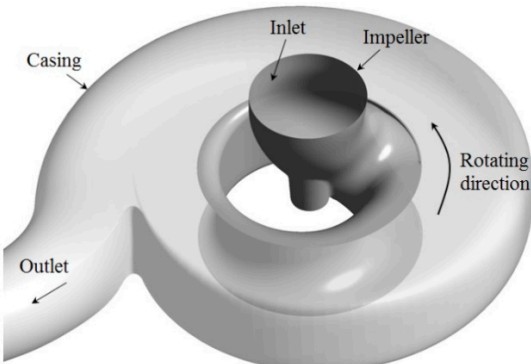

**Figure 1.** Geometry of single-channel pump.

As shown in Figure 2, the cross-sectional area of the impeller of the reference model was designed considering the minimum thickness in the range 0–70° of the impeller. The impeller was designed to have the maximum cross-sectional area at the 360°. In the case of the volute, as shown in Figure 2, the cross-sectional area was designed to constantly increase from 0 to 360°. A more detailed description can be referred to in the authors' previous work.

The head and flow coefficients, defined as below, for the reference pump are 0.074 and 0.019, respectively.

$$\psi = \frac{gH}{N^2 D^2},$$

(1)

$$\phi = \frac{Q}{ND^3},$$

(2)

where, $N$, $D$, $g$, $H$ and $Q$ are the rotational speed, impeller diameter, gravity acceleration, total head and volume flow rate, respectively. The additional design information is listed in Table 1 [23].

**Table 1.** Design major specifications of the single-channel pump [23].

| | |
|---|---|
| Flow coefficient ($\phi$) | 0.019 |
| Head coefficient ($\psi$) | 0.074 |
| Rotational speed (RPM) | 1760 |
| Impeller inlet-outlet diameter ratio | 1.9 |

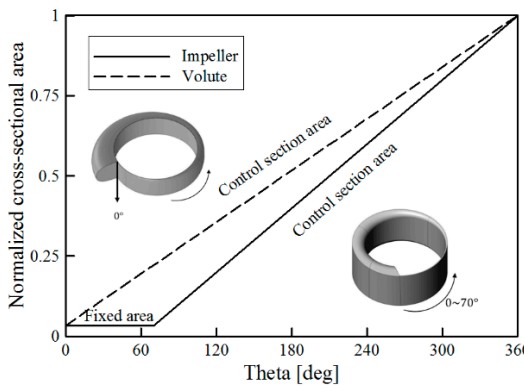

**Figure 2.** Cross-sectional area distributions along the theta angle of impeller and volute.

### 2.2. Numerical Analysis

Commercial code ANSYS CFX-15.0 was employed for the hydraulic analysis of the single-channel pump. For turbulence closure, the k-ω based shear stress transport (SST) model [25] was applied and three-dimensional steady and unsteady incompressible RANS equations were solved. Generally, the SST model combines a k-ω model of the near-wall flow and k-ε model of the bulk flow. A blending function was used to connect these two models. The SST model is well known to predict precisely flow separations under adverse pressure gradients.

The 3-D model for the volute and impeller was generated by Solidworks 2015 and the computational grids system for numerical analysis was created by ICEM-CFD. Since the computational domain includes both stationary and rotating domains, the specific processing technique provided by ANSYS CFX-15.0 was used at the interface. The stage (or mixing plane) and the transient Rotor-Stator interface methods were used at the interface between the rotating and stationary domains in the RANS and unsteady RANS analyses, respectively.

Freshwater was employed for working fluid and the inlet boundary condition of the pump was set to atmospheric pressure. The surfaces of the volute and impeller are considered as adiabatic and no-slip conditions. The outlet boundary condition was set to the mass flow rate, and the numerical analysis was performed while changing the mass flow rate.

The grid system consists of a tetrahedral grid in the volute and impeller domains. In this study, $y^+$ was kept below 2 to use the low Reynolds SST model, so that the flow near the wall was accurately analyzed. To eliminate the grid dependency of the numerical solutions, grid dependency test was performed at the design mass flow rate for the hydraulic performance of the pump. Based on the

results in the previous work [23], the optimal grids system (a total of 2,500,000 nodes; 1,300,000 nodes in the impeller domain and 1,200,000 nodes in the volute domain) was selected, as shown in Figure 3.

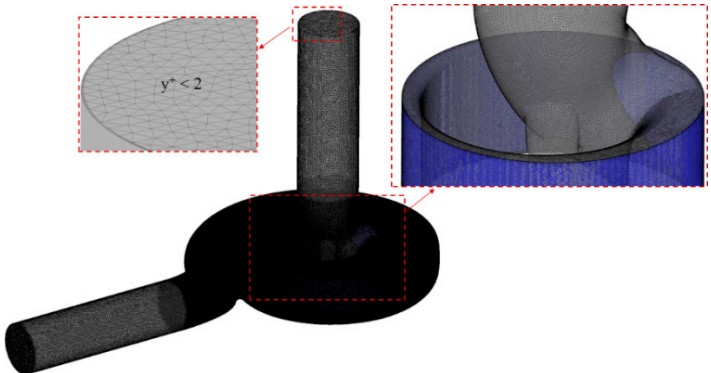

**Figure 3.** Grid systems of computational domain.

To determine the convergence of the numerical calculations, the root-mean-square (RMS) values of the residuals of the governing equations were set to be less than $10^{-5}$. The calculation for the steady RANS analysis were performed using an Intel Xeon CPU (2.70 GHz) processor and the CPU running time for each analysis was about 4 h.

Steady RANS results were employed as initial values for unsteady RANS analysis, and coefficient loop with time scale control and time step were set to three times and 0.000947 s, respectively. Unsteady RANS analysis results were obtained after five revolutions (180 iterations with a total time of 0.01704775 s), and the CPU running time for each analysis was approximately 8 h. Through unsteady RANS analysis, characteristics of radial force sources due to flow interaction at the interface between the rotating and stationary domains were obtained and analyzed.

## 3. Optimization Techniques

The flowchart for the multi-objective optimization process using surrogate modeling is shown in Figure 4. First, the objective functions and constraints are defined according to the design goal. Subsequently, the design variables considering constraints are chosen. The LHS [20] is used as DOE to select the design points and the objective functions are evaluated by (U)RANS analysis at these design points. The next step is the construction of surrogate model to approximate the objective functions and a searching algorithm, that is, hybrid PSO-GA, is used to combine and formulate these surrogate models. Finally, Pareto-optimal solutions—that is, a collection of non-dominated solutions—are derived and the multi-objective optimization procedure is terminated [26].

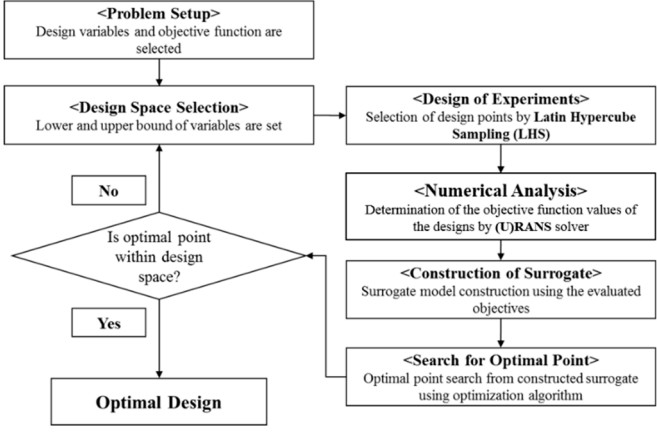

**Figure 4.** Optimization procedure.

### 3.1. Optimization Goal

The design goals of the current multiple optimization were to concurrently reduce the radial force sources and improve the hydraulic efficiency. The hydraulic efficiency, one of the objective functions, is defined as follows:

$$F_{\eta} = -\frac{\rho g H Q}{P},$$ (3)

where, $H$, $Q$, $\rho$, $g$, and $P$ represent total head, volume flow rate, the density, gravity acceleration, and power, respectively.

Other objective functions for the radial force sources are the distance of the mass center of the sweep area from the origin and the sweep area of the radial force during one revolution, as follows:

$$A_s = \frac{1}{2}\sum_{i=0}^{n-1}(x_i y_{i+1} - x_{i+1} y_i),$$ (4)

where, $A_s$ represents the signed area of the polygon, considered as the sweep area of the radial force during one revolution. The centroid of a non-self-intersecting closed polygon defined by $n$ vertices $(x_0, y_0)$, $(x_1, y_1)$, ... , $(x_{(n-1)}, y_{(n-1)})$ is defined as the point $(C_x, C_y)$, as follows:

$$C_x = \frac{1}{6A_s}\sum_{i=0}^{n-1}(x_i + x_{i+1})(x_i y_{i+1} - x_{i+1} y_i),$$ (5)

$$C_y = \frac{1}{6A_s}\sum_{i=0}^{n-1}(y_i + y_{i+1})(x_i y_{i+1} - x_{i+1} y_i).$$ (6)

The vertices in these equations are defined as being numbered in sequence along the perimeter of the polygon. Hence, the distance of the mass center of the sweep area from the origin can be finally defined as follows:

$$F_{radial} = \sqrt{C_x^2 + C_y^2}.$$ (7)

Thus, in order to fulfil the design goals described above, multi-objective optimization was performed using $F_\eta$ and $F_{radial}$ as objective functions.

In this study, five geometric parameters associated to the internal flow characteristics of the pump were chosen as design variables to control the cross-sectional area of the volute and impeller. The distribution of the cross-sectional area of the internal flow of impeller and volute can be controlled smoothly by adjusting the control points represented by a third and fourth order Bezier-curve, respectively, as shown in Figure 5. Hence, the variations in the y-axes for two control points (CP$_1$ and CP$_2$) for the impeller and three control points (CP$_3$, CP$_4$ and CP$_5$) for the volute were selected. Figure 5 shows the defined design variables and ranges are listed in Table 2.

**Table 2.** Ranges of design variables.

|  | LB | Ref. | UB |
|---|---|---|---|
| CP 1 | 0.03 | 0.32 | 0.61 |
| CP 2 | 0.42 | 0.71 | 1.00 |
| CP 3 | 0.00 | 0.25 | 0.50 |
| CP 4 | 0.00 | 0.50 | 1.00 |
| CP 5 | 0.50 | 0.75 | 1.00 |

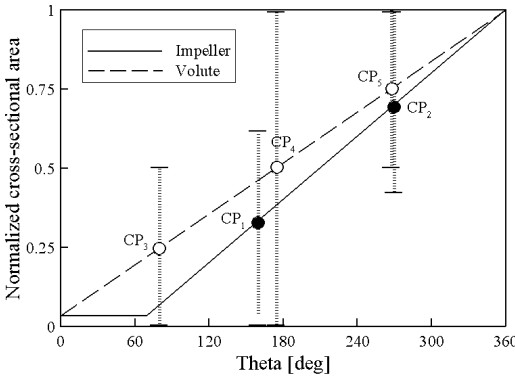

**Figure 5.** Defined design variables.

### 3.2. Surrogate Modeling

In the present work, 54 design points were generated for the five design variables by LHS using the MATLAB function 'lhsdesign' [27]. LHS is an effective sampling technique that uses an $m \times n$ simulation matrix where $m$ is the number of levels (sampling points) to be examined and $n$ is the number of design parameters. Each of the $n$ columns of the matrix containing the levels, 1, 2, ... , $m$, is randomly paired to form a Latin hypercube. This approach produces random sample points, ensuring that all portions of the design space are represented. The objective function values at these design points were calculated by (U)RANS analysis, as shown in Table A1 (Appendix A).

The RBNN model was applied for the surrogate modeling from the objective functions. The RBNN as a neural network model, presented in Figure 6, consists of a hierarchical structure that starts at the input stage and ends at the output stage, where each stage is known as a layer. There are intermediate layers between the input and output stages, known as hidden layers. By controlling the number and size of these hidden layers, arbitrary complex functions can be learned, and the surrogate models can be constructed. The RBNN includes a two-layered hidden network containing a layer of radial neurons and an output of linear neurons. The hidden layer uses a series of radial primitives to nonlinearly modify the input space to the intermediate space. The output layer then executes a linear combiner to produce the desired target. In this study, MATLAB built-in code 'radbas' and 'newrb' were used to perform radial basis function-based machine learning [27]. 'Newrb' makes a two-layer network. As mentioned above, the first layer constructed by 'radbas' is radial basis neuron network and the second layer is linear neuron network. This combination model $f(\mathrm{x})$ for the formulation can be represented as a linear combination of a set of N radially-symmetric functions as follow:

$$f(\mathrm{x}) = \sum_{j=1}^{N} w_j \varnothing_j, \tag{8}$$

where $w_j$ is the weight and $\Phi_j$ is radial basis function. Machine learning finds the input vector with the largest error in the network and adjusts the weight or neurons of this vector to minimize the error in the network. This process is repeated until the network's mean squared error falls below the set goal. The construction process can be accessed through graphical user interface (GUI) environment using the application 'neural net fitting', which is supported within MATLAB [27].

In order to construct the RBNN model, an initial value must be assigned to the spread constant (SC), that is, a constant for the first hidden layer (nonlinear function) of the neural network. Because machine learning is based on this initial SC value, it should be chosen carefully.

The network training was implemented by changing the SC values to identify the minimum cross-validation error. The parameters $SC_1$ and $SC_2$ are constants that are utilized to configure the surrogate model with respect to the efficiency and radial force, respectively. The SC values were chosen using a k-fold cross-validation test [28], which is a validation method that estimates how the results of

statistical analysis are generalized to independent datasets. The process of the k-fold cross-validation test used in the present work is described as follows:

Step 1  Construct a surrogate model using the 53 experimental points, except for one point of the 54 experimental points.

Step 2  Compare the value of the objective function at the location of the experimental point excluded from Step 1 (between CFD simulation value and predicted value by the surrogate model).

Step 3  This process is carried out at all experimental points. Then, evaluate the sum of the errors between predicted and CFD simulation values.

As shown in Figure 7, the error was minimized when the values of $SC_1$ and $SC_2$ were 0.2 and 1.0, respectively. Neural networks with various numbers of neurons were trained 10 times and the averaged adjusted $R^2$ values are compared (Figure 8). The same quantity of neurons was used in each layer; when using 12 neurons, the statistically most accurate models with adjusted $R^2$ equals to 1.00 and 0.78 could be achieved.

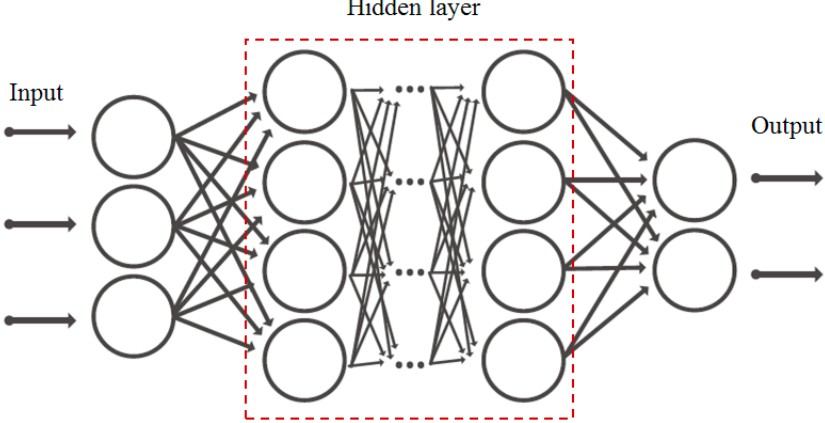

**Figure 6.** Schematic of neural network.

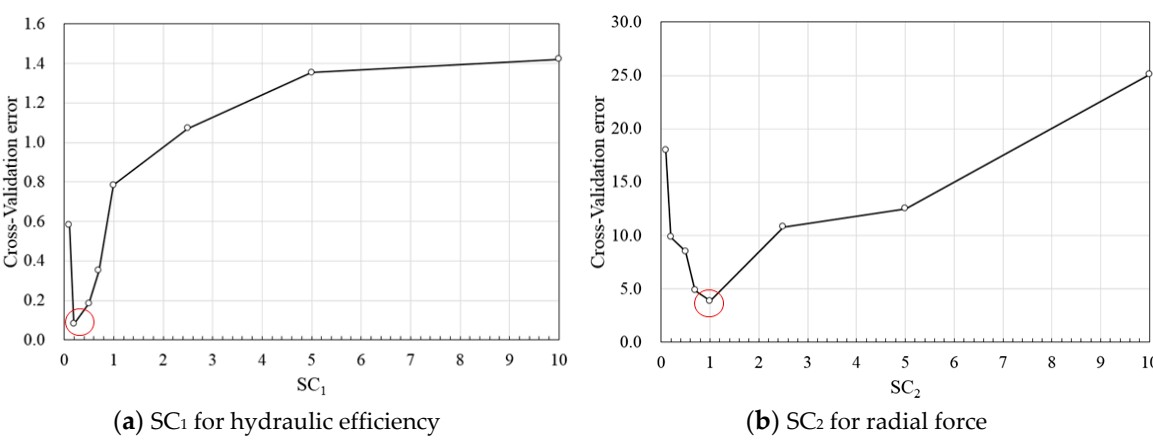

(**a**) $SC_1$ for hydraulic efficiency  (**b**) $SC_2$ for radial force

**Figure 7.** Cross-validation error vs. Spread constant (SC) value.

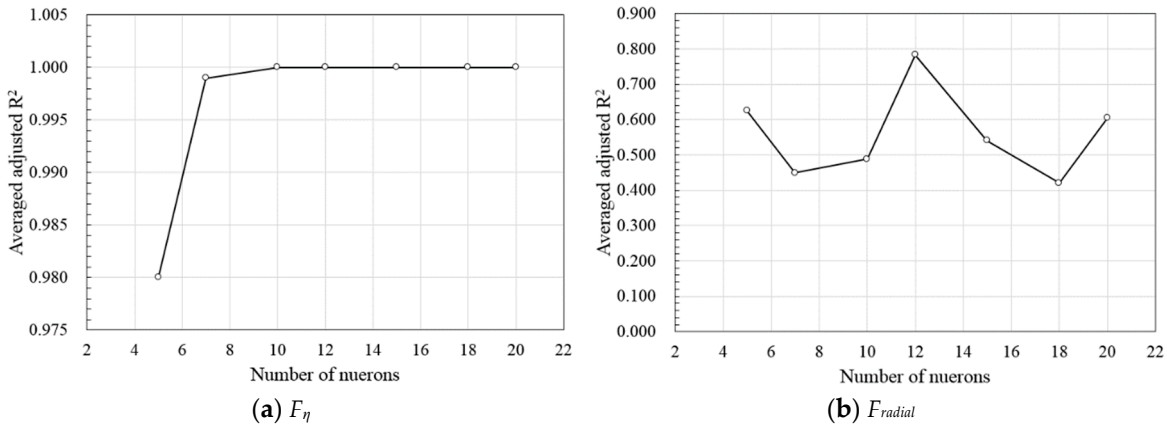

**(a)** $F_\eta$　　　　　　　　　　　　　　　**(b)** $F_{radial}$

**Figure 8.** Comparison of averaged adjusted $R^2$ of RBNN.

### 3.3. Searching Algorithm

Once the surrogate model is constructed, it is necessary to find the optimal point on the fitness assignment, which is the output in the form of a function of the surrogate model. In the present work, a hybrid PSO-GA algorithm that was most recently developed [18] was used as searching algorithm to identify a global optimal solution within the fitness assignment. It is known to have a high reliability. Hybrid strategies combine two or more algorithms to maximize the benefits of each algorithm. One benefit of the PSO algorithm is its simplicity and another one is that the PSO algorithm controls the convergence criteria to identify the optimal solution in a short time. The GA, on the other hand, has the advantage of finding the global optimal solution by controlling the genetic search operators—that is, the crossover and mutation rates—which affect the convergence. The locations of the global best particles are updated by the genetic search operators to prevent premature convergence. By using the genetic search operators, the locations of the particles can be exchanged among the particles and thus have the ability to escape the local optima.

The procedure of the hybrid PSO-GA algorithm is represented in Figure 9 [15]. This algorithm shares the initial number of particles of the PSO with the number of populations of the GA. The first step of this procedure is to identify an initial optimal solution using the particles of the PSO. The next step is to modify the locations of the particles using the genetic search operators and identify the improved optimal solution. The procedure is competed when the termination criteria are met by referring to the updated location; otherwise, it is repeated. In the present work, the following parameters and termination criteria were used: population size (i.e., number of particles) = 400, crossover fraction = 0.35, mutation fraction = 0.70, generations = 1000, function tolerance = $10^{-6}$, and initial velocity of particles = 2.5. The pseudo-code of the hybrid PSO-GA used in this study is shown in Table A2 (Appendix B).

### 3.4. Optimization Results

For the optimization, hybrid PSO-GA with RBNN was used to search the Pareto-optimal front in a feasible solution space bounded by the lower and upper bounds of the design variables. Pareto-optimal solutions with the optimum trade-off between the two conflicting objective functions are plotted in Figure 10.

The arbitrary optimum design (AOD) was obtained as the final design near the center of the Pareto-optimal front, to improve the prediction of both the objective functions compared to the reference. The values of the design variables and objective functions of the AOD, along with those for the reference design, are listed in Table 3. The AOD showed an improvement of 3.07% and 44.80% for the hydraulic efficiency and radial force, respectively, compared with the reference design. In contrast, the relative errors between (U)RANS and the predicted values for these objective functions were 0.16% and 8.77%, respectively.

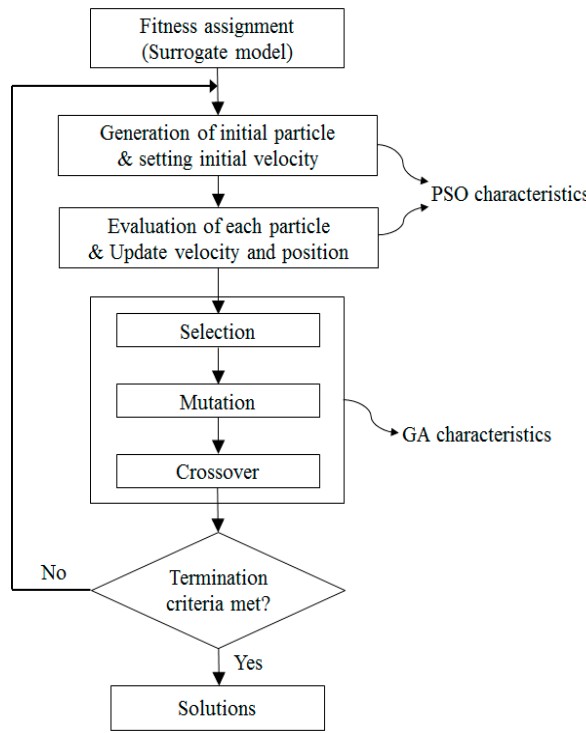

**Figure 9.** Algorithm of hybrid PSO-GA.

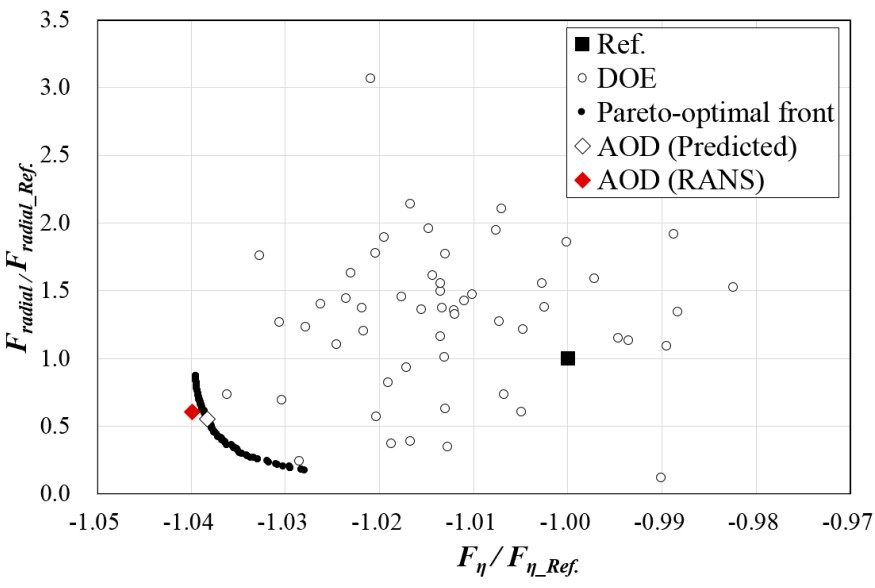

**Figure 10.** Pareto-optimal solutions.

**Table 3.** Results of optimization.

| | Design Variables | | | | | Predicted Values | | (U)RANS | | Relative Error (%) | |
|---|---|---|---|---|---|---|---|---|---|---|---|
| | CP 1 | CP 2 | CP 3 | CP 4 | CP 5 | $F_\eta/F\eta\_Ref.$ | $F_{radial}/F_{radial\_Ref}$ | $F_\eta/F_{\eta\_Ref.}$ | $F_{radial}/F_{radial\_Ref}$ | $F_\eta/F_{\eta\_Ref.}$ | $F_{radial}/F_{radial\_Ref}$ |
| **AOD** | 0.600 | 0.004 | 0.003 | 0.214 | 0.837 | −1.038 | 0.5520 | −1.040 | 0.6050 | 0.16 | 8.77 |

## 4. Results and Discussion

### 4.1. Unsteady Analyses of Internal Flow Field

In a previous work of the authors [29], the accuracy of the numerical analysis used in this work was proved in comparison with the experimental result and then the results of the numerical analysis were consequentially considered valid.

To quantitatively verify the improvement in hydraulic performance of the AOD owing to the optimization, the internal flow paths for the reference model and AOD were compared systematically with their cross-sectional area distributions, as shown in Figures 11–17. Figure 11 shows the cross-sectional area distributions in the single-channel impeller and volute for the reference model and AOD. The cross-sectional area distribution of the flow path of the impeller in AOD most closely resembles a concave shape, as shown in Figure 11a. This implies that the flow path in the impeller of AOD was narrowed slightly compared with the reference model. The cross-sectional area distribution in the region of the volute upstream (from 0 to 240°) in AOD also decreased considerably, while the cross-sectional area distribution in the downstream region after 240° increased slightly compared with the reference model, as shown in Figure 11b. The concave flow path in the region of the impeller downstream and volute upstream was found to be considerably sensitive for the high efficiency and low radial force design of the single-channel pump. As a result, it can be concluded that the single-channel impeller and volute shapes should be optimized concurrently to design a single-channel pump with high hydraulic efficiency and low radial forces.

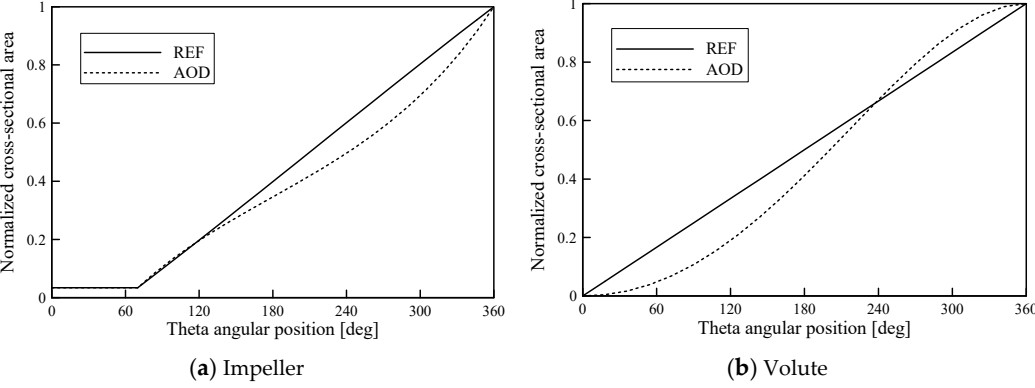

(**a**) Impeller  (**b**) Volute

**Figure 11.** Cross-sectional area distributions of the flow path.

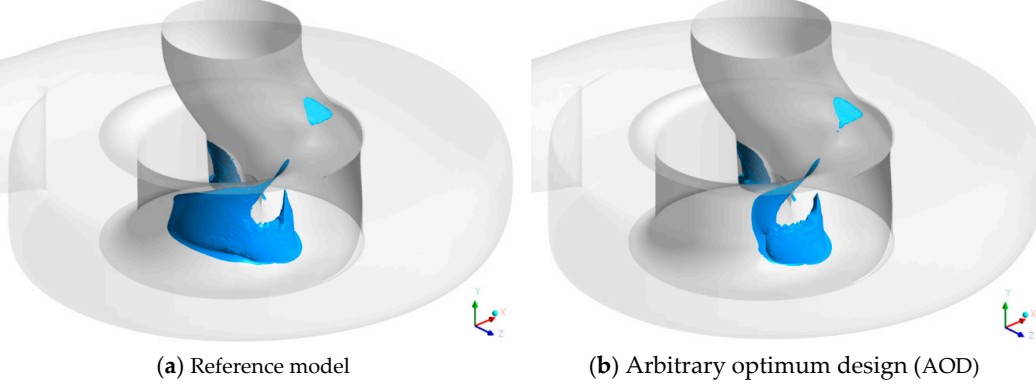

(**a**) Reference model  (**b**) Arbitrary optimum design (AOD)

**Figure 12.** Time-averaged iso-surfaced contours at a low velocity of 1.5 m/s.

Figure 12 shows time-averaged iso-surfaced contours having a low velocity component (1.5 m/s), which is the flow separation zone, for the reference model and AOD. For the reference impeller, the large flow separation zone formed along with the internal annulus wall in the single-channel

impeller, as shown in Figure 12a. On the contrary, the flow separation zone was reduced remarkably in the downstream region of the optimized impeller, as shown in Figure 12b.

Figure 13 shows the time-averaged streamline distributions in the impeller surface wall of the reference and optimized models. Extreme flow separation was occurred in the hub wall by abruptly changing the flow path for both models. However, the limited streamline in the AOD was moved from the impeller outside wall to the inside wall by the optimized flow path. It can be seen that these changing flows contributed to the decrease in flow separation and increase in the hydraulic efficiency owing to optimization. Thus, the decrease in the flow separation from the impeller can be made by the smooth pressure increase by matching with the optimized volute shape, as shown in the time-averaged pressure contours on the mid-span in Figure 14. These results illustrate the improvement in the entire hydraulic efficiency of the pump.

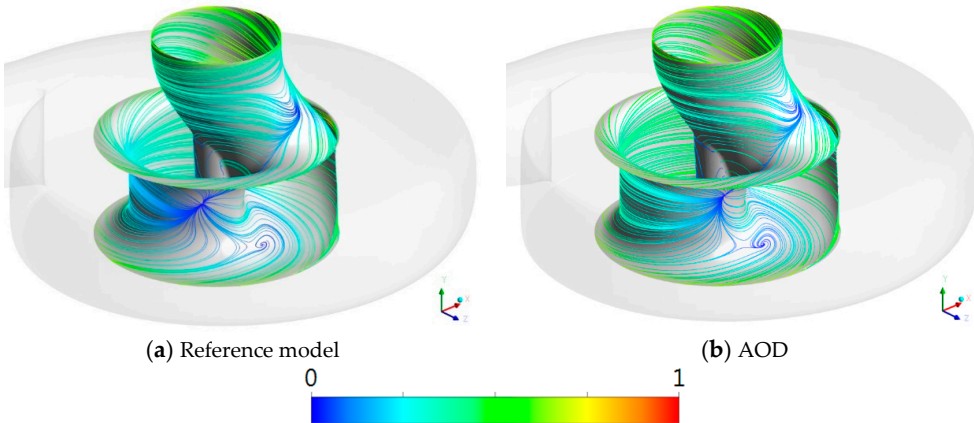

(**a**) Reference model                    (**b**) AOD

0                                          1

**Figure 13.** Time-averaged pressure contours on impeller surface wall.

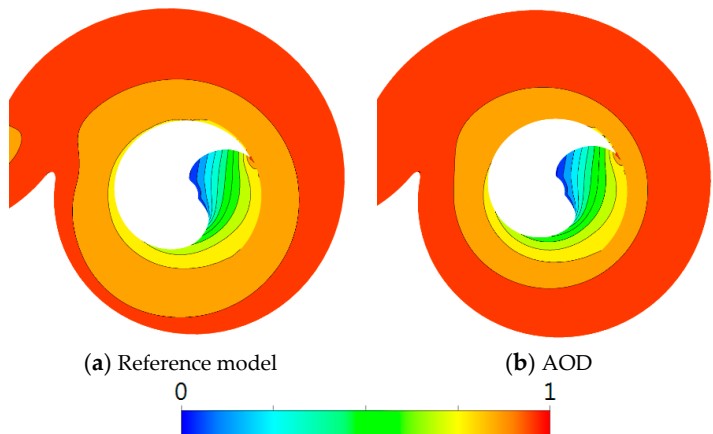

(**a**) Reference model                    (**b**) AOD

0                                          1

**Figure 14.** Time-averaged pressure contours at mid-span.

The sweep area distributions generated by the unsteady radial force components during the impeller one rotation for the reference model and AOD are plotted in Figure 15; they are averaged at the cylindrical surface near exit of the single-channel impeller. Here, all $F_x$ and $F_y$ values of the radial force were normalized based on the maximum radial force of the reference design. As shown in Figure 15, the sweep area distribution in the reference design leaned a little toward the fourth quadrant, whereas it was closer to the origin for AOD. Moreover, the sweep area in AOD reduced significantly compared with the reference model. As discussed previously in Section 3.4, the radial force in AOD decreased by 14.73, compared with the reference design.

Figure 16 shows the unsteady fluctuation distributions in the net radial forces during the impeller one rotation in the reference model and AOD. Here, the net radial force values were normalized using the maximum net radial force value of the reference design. The fluctuation amplitude in AOD

generally decreased during one revolution of the impeller, especially for flat formed angles from 0 to 240°. In addition, the net radial force distribution in AOD was lower and mostly flat compared with the normalized values of the reference model. The remarkable reduction in the radial force sources can be explained clearly by the optimization of this work.

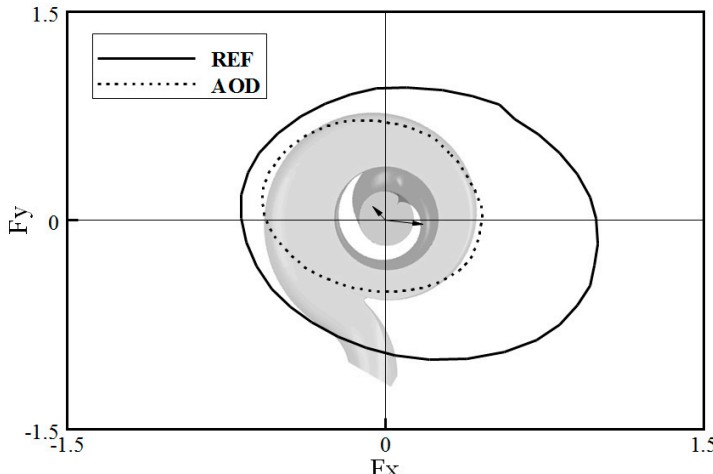

**Figure 15.** Sweep area distributions during one revolution of impeller.

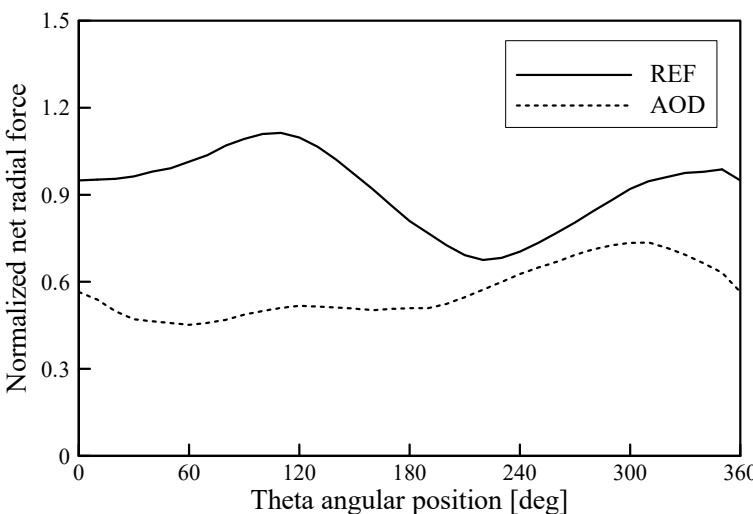

**Figure 16.** Unsteady net radial force fluctuations during one rotation of impeller.

The instantaneous change in the unsteady pressure values on the cylindrical surface near the exit of the impeller for the reference model and AOD are shown in Figure 17. Here, the impeller one rotation is divided into five steps to clearly explain the variation in the flow phenomenon with time during impeller one rotation. In the reference model, the pressure variation caused by the impeller–volute interaction is prominent and then increases gradually, especially in Figure 17b. As discussed previously in Figure 15, the high-pressure zone is the fourth quadrant where the sweep area distribution leans slightly. Owing to this repetitive rotation, fluid-induced vibration caused by unsteady radial forces are observed. These unbalancing phenomena have an adverse effect on pump performance. In contrast, unsteady pressure distribution is generally uniform during the impeller one rotation; especially, at the same time of the reference model. With these trends, the high-pressure variation induced by the impeller–volute interaction clearly disappeared in Figure 17. Therefore, the AOD resulted in mostly stable flows in the pump with high efficiency and low radial forces. There is a remarkable decrement in the fluid-induced vibration induced by the impeller–volute interaction from the current multiple optimization result.

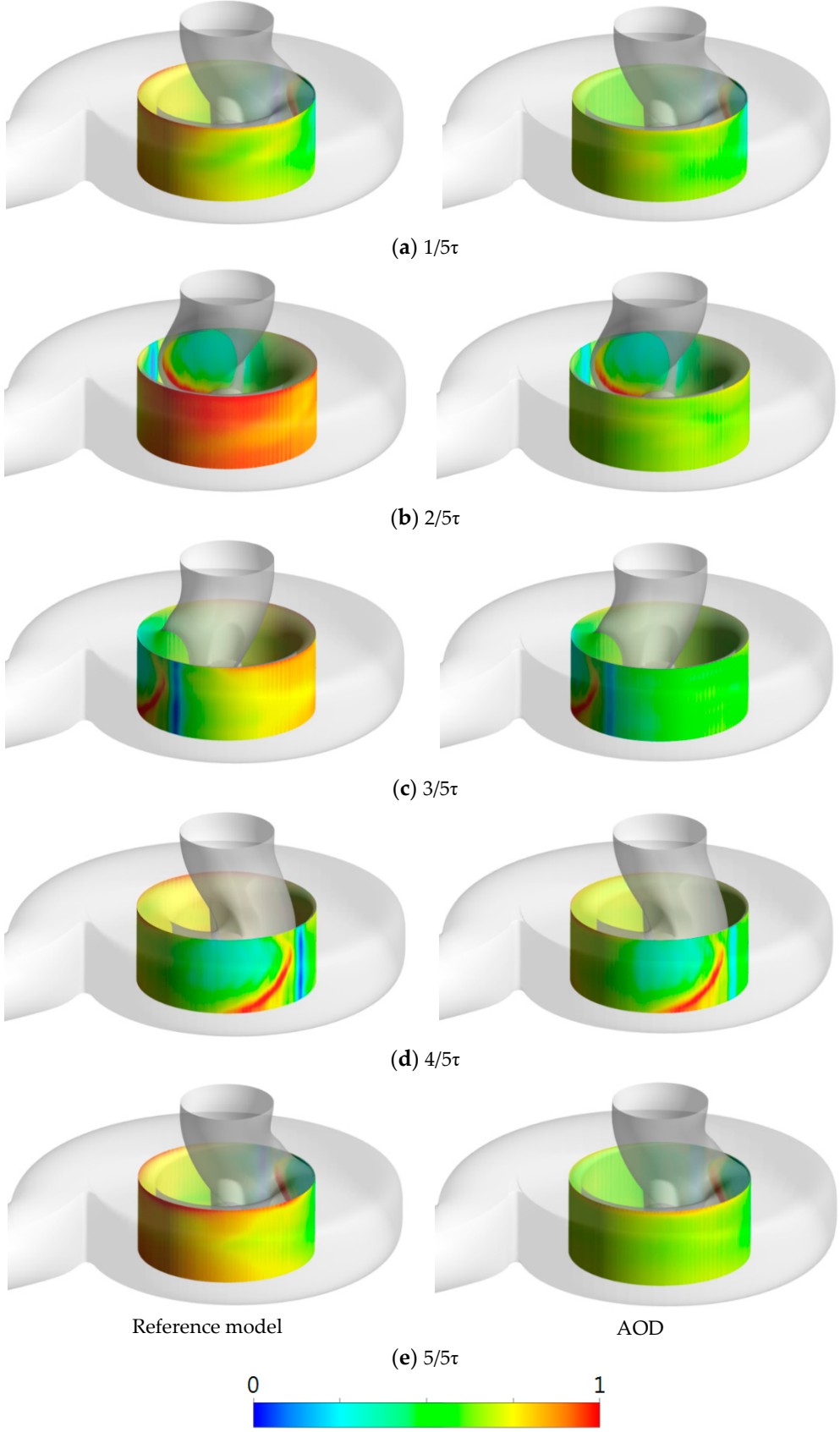

(**a**) 1/5τ

(**b**) 2/5τ

(**c**) 3/5τ

(**d**) 4/5τ

Reference model                                                                                     AOD

(**e**) 5/5τ

0                                                                                     1

**Figure 17.** Instantaneously changed unsteady pressure contours during one rotation at exit surface of impeller.

Figure 18 shows the spectra of the normalized amplitude values based on the wall pressure fluctuations measured at the observation point on the casing wall near the tongue of the volute located in the fourth quadrant in Figure 15. In Figure 18, the first blade passing frequency (BPF) is measured at approximate 30 Hz, and this is represented by the fluid-induced vibration characteristics from the impeller–volute interaction. These peak amplitude values are clarified at every harmonic BPF in 30 Hz steps due to the periodic motion of impeller rotation. In the AOD, a remarkable decrement in the amplitude values at the first BPF was occurred, and the decrement in fluid-induced vibration induced by the impeller–volute interaction was confirmed.

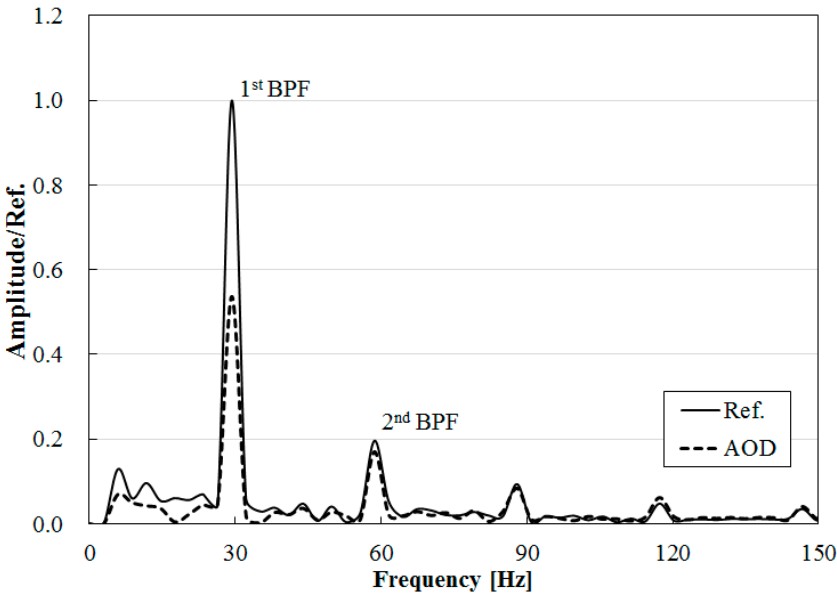

**Figure 18.** Spectra of amplitude values on casing wall near the tongue of volute.

*4.2. Performance Verification of the Optimized Prototype Model*

To confirm the hydraulic performance of the optimized model obtained through the numerical optimization technique, the prototype model was manufactured in Figure 19. Figure 20 shows the schematic of the experimental apparatus constructed in Korea Institute of Industrial Technology to measure the hydraulic head and efficiency of the prototype model. The experimental apparatus was constructed based on the standards of KS B 6301 and 6321. In addition, the measuring devices for the pressure sensor, temperature sensor, magnetic flow meter, and digital power meter were calibrated from the qualified office of Korea. Figure 21 shows the experimental results for the head and efficiency of the prototype model, with the numerical analysis results. As shown in Figure 21, the head coefficient curves of the experimental and numerical results are similar in the entire range, the head value, in particular, is almost the same at the design flow coefficient. In contrast, the efficiency curves of the numerical analysis including the motor efficiency are relatively higher than those of the experimental results, because the numerical results do not include the mechanical losses. However, the general curve trend matched well. From these results, the required single-channel pump could be designed through the proposed numerical optimization technique; hence, the technique was demonstrated to be feasible.

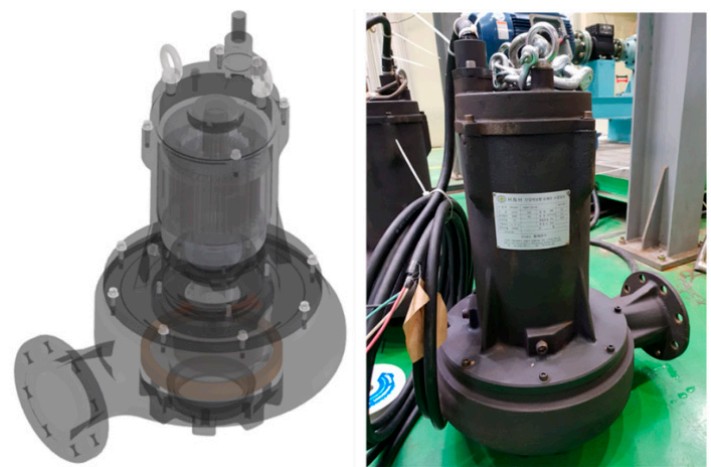

**Figure 19.** Prototype model for experimental test.

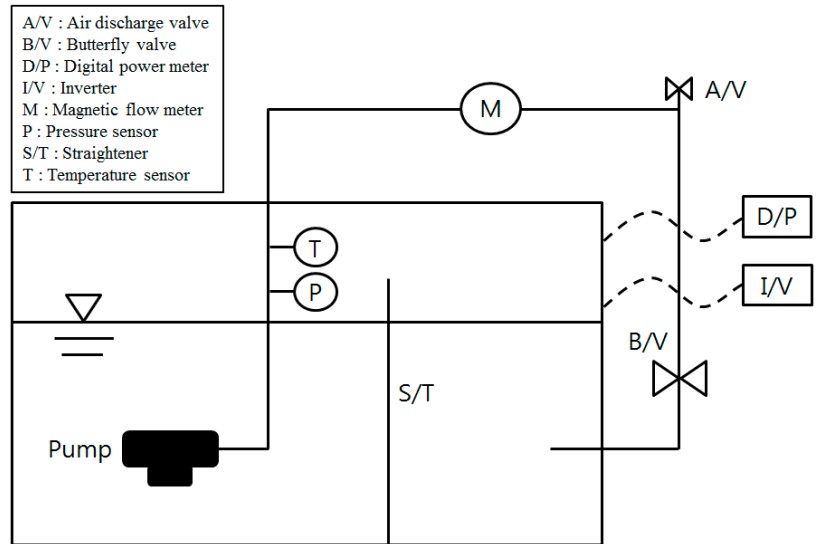

**Figure 20.** Experimental apparatus for pump test.

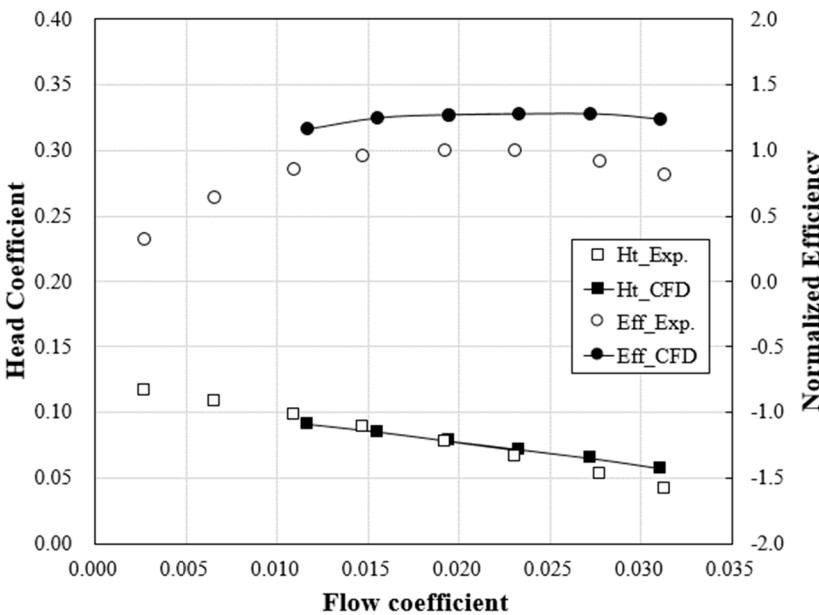

**Figure 21.** Verification results of prototype model.

## 5. Conclusions

In this study, a multiple optimization technique using hybrid PSO-GA was introduced to effectively enhance the efficiency and radial force sources with fluid-induced vibration of the single-channel pump. 3-D steady and unsteady RANS analyses were performed to investigate the hydraulic characteristics of the single-channel pump using ANSYS CFX-15.0. As the design variables, design parameters for controlling the internal cross-sectional areas of the single-channel impeller and volute were chosen in the authors' previous study. Two performance functions, i.e., hydraulic efficiency and radial force, were defined as the objective functions. The optimization based on the surrogate model was performed considering reasonable calculation time. The RBNN method was used as a surrogate model, and it was repeatedly constructed 10 times with a different number of neurons constituting the layer for statistical analysis. As a result, when 12 neurons were used, the most accurate surrogate model was constructed. Through optimization with RBNN consisting of 12 neurons, the hydraulic efficiency of the optimum design increased by 3.07%, and the radial force caused by impeller–volute interaction reduced by 44.80%, compared to the reference design.

In the optimum design, the flow path in the impeller was narrow in comparison with the reference design. The flow path in the volute upstream of the optimum design was narrowed, while that in the volute downstream was slightly increased, in comparison with the reference design. These structural changes balanced the pressure distribution along the impeller–volute interface and delayed the flow separation on the impeller surface. As a result, it was confirmed through (U)RANS analysis that the fluid-induced vibration of the optimum design reduced in comparison with the reference design. The optimum design obtained in the present study was used to manufacture a prototype model, and the hydraulic performances, i.e., efficiency and head, were verified experimentally. Thus, reasonable results were obtained with the required design specification. However, the precise vibration test was not conducted, and only the vibration values were measured briefly using the simple vibration meter. This is because it was very difficult to simultaneously measure detailed vibration with the hydraulic performances in the experimental apparatus, and there was no submersible three-axis vibration sensor. Therefore, in a future work, the vibration characteristics of the single-channel pump will be investigated in more detail using a three-axis vibration meter, and the flow characteristics including solid wastes will be studied.

**Author Contributions:** Conceptualization, J.-H.K., S.K. and Y.-S.C.; methodology, S.-B.M.; software, J.-H.K., S.K. and S.-B.M.; validation, J.-H.K., S.K. and Y.-S.C.; formal analysis, J.-H.K. and S.-B.M.; investigation, J.-H.K., S.K. and S.-B.M.; writing original draft preparation, J.-H.K. and S.-B.M.; supervision, J.-H.K. and K.-Y.K.; project administration, J.-H.K; funding acquisition, J.-H.K;

**Funding:** This work was supported by the Demand-based-Manufacturing Technique Commercialization R&D Project of the Korea Institute of Industrial Technology (KITECH) (No. JB190004), which was funded by the Ministry of Science and ICT (MSIT). The authors gratefully acknowledge this support.

**Conflicts of Interest:** The author(s) declared no potential conflicts of interest with respect to the research, authorship, and/or publication of this article.

## Nomenclature

| AOD | Arbitrary optimum design |
| --- | --- |
| BPF | Blade passing frequency |
| CFD | Computational fluid dynamics |
| CP | Control point |
| D | Diameter of impeller |
| DOE | Design of experiments |
| FSI | Fluid-structure interaction |
| g | Gravity acceleration |
| GA | Genetic algorithm |
| H | Total head |

| | |
|---|---|
| LDV | Laser Doppler velocimetry |
| LHS | Latin hypercube sampling |
| N | Rotational speed |
| P | Power |
| PSO | Particle swarm optimization |
| Q | Volume flow rate |
| RANS | Reynolds-averaged Navier-Stokes |
| RBNN | Radial basis neural network |
| SC | Spread constant |
| SST | Shear stress transport |
| (U)RANS | Unsteady Reynolds-averaged Navier-Stokes |
| $\rho$ | Density |
| $\phi$ | Flow coefficient |
| $\Psi$ | Head coefficient |

## Appendix A

**Table A1.** Design variables and objective functions values at LHS design points.

| | CP 1 | CP 2 | CP 3 | CP 4 | CP 5 | $F_\eta/F_{\eta\_Ref.}$ | $F_{radial}/F_{radial\_Ref}$ |
|---|---|---|---|---|---|---|---|
| 1 | 0.000 | 0.000 | 1.000 | 0.000 | 0.000 | −1.021 | 3.064 |
| 2 | 0.000 | 0.000 | 0.000 | 0.000 | 1.000 | −1.033 | 1.759 |
| 3 | 0.000 | 0.000 | 1.000 | 1.000 | 0.000 | −1.017 | 2.144 |
| 4 | 1.000 | 1.000 | 0.000 | 0.000 | 1.000 | −0.982 | 1.527 |
| 5 | 0.265 | 0.306 | 0.429 | 0.980 | 0.837 | −1.026 | 1.403 |
| 6 | 1.000 | 0.469 | 0.286 | 0.592 | 0.571 | −0.994 | 1.133 |
| 7 | 0.551 | 0.735 | 0.327 | 0.490 | 0.000 | −1.012 | 1.355 |
| 8 | 0.959 | 0.184 | 0.857 | 0.531 | 0.327 | −1.007 | 2.109 |
| 9 | 0.020 | 0.857 | 0.224 | 0.449 | 0.265 | −1.014 | 1.161 |
| 10 | 0.939 | 0.653 | 0.796 | 0.673 | 0.878 | −0.988 | 1.344 |
| 11 | 0.449 | 0.510 | 0.735 | 0.000 | 0.469 | −1.016 | 1.362 |
| 12 | 0.347 | 0.408 | 0.776 | 0.918 | 0.163 | −1.015 | 1.958 |
| 13 | 0.694 | 0.041 | 0.510 | 0.694 | 0.694 | −1.024 | 1.441 |
| 14 | 0.633 | 0.878 | 0.122 | 0.735 | 0.633 | −1.003 | 1.377 |
| 15 | 0.796 | 0.714 | 0.592 | 0.061 | 0.082 | −1.000 | 1.861 |
| 16 | 0.571 | 0.265 | 1.000 | 0.388 | 0.714 | −1.020 | 1.778 |
| 17 | 0.857 | 0.245 | 0.061 | 0.184 | 0.898 | −1.013 | 0.628 |
| 18 | 0.918 | 0.694 | 0.449 | 0.122 | 0.776 | −1.013 | 0.347 |
| 19 | 0.388 | 0.959 | 0.551 | 0.265 | 0.408 | −1.011 | 1.429 |
| 20 | 0.245 | 0.918 | 0.980 | 0.755 | 0.245 | −1.014 | 1.495 |
| 21 | 0.184 | 0.000 | 0.184 | 0.551 | 0.755 | −1.036 | 0.738 |
| 22 | 0.306 | 0.020 | 0.878 | 0.857 | 0.551 | −1.023 | 1.630 |
| 23 | 0.837 | 0.061 | 0.633 | 0.143 | 0.673 | −1.013 | 1.011 |
| 24 | 0.327 | 0.429 | 0.020 | 0.510 | 0.306 | −1.020 | 0.569 |
| 25 | 0.878 | 0.980 | 0.245 | 0.327 | 0.367 | −0.990 | 0.122 |
| 26 | 0.469 | 0.939 | 0.673 | 0.837 | 0.816 | −1.003 | 1.557 |
| 27 | 0.429 | 0.796 | 0.000 | 0.224 | 0.612 | −1.029 | 0.244 |
| 28 | 0.898 | 0.571 | 0.694 | 0.939 | 0.347 | −0.997 | 1.591 |
| 29 | 0.816 | 0.837 | 0.918 | 0.306 | 0.490 | −0.989 | 1.920 |
| 30 | 0.714 | 0.143 | 0.367 | 0.653 | 0.143 | −1.013 | 1.559 |
| 31 | 0.143 | 0.082 | 0.653 | 0.082 | 0.592 | −1.031 | 1.271 |
| 32 | 0.735 | 0.592 | 0.898 | 0.612 | 0.020 | −1.008 | 1.950 |
| 33 | 0.286 | 0.776 | 0.959 | 0.286 | 0.796 | −1.007 | 1.275 |
| 34 | 0.224 | 0.327 | 0.265 | 0.041 | 0.102 | −1.025 | 1.103 |
| 35 | 0.061 | 0.286 | 0.306 | 0.878 | 0.204 | −1.022 | 1.207 |
| 36 | 0.755 | 0.490 | 0.041 | 0.163 | 0.184 | −1.007 | 0.738 |

**Table A1.** *Cont.*

|     | CP 1  | CP 2  | CP 3  | CP 4  | CP 5  | $F_\eta/F_{\eta\_Ref.}$ | $F_{radial}/F_{radial\_Ref}$ |
| --- | ----- | ----- | ----- | ----- | ----- | ------- | ------ |
| 37 | 0.102 | 0.633 | 0.143 | 0.633 | 0.857 | −1.019 | 0.373 |
| 38 | 0.673 | 1.000 | 0.612 | 0.776 | 0.286 | −0.995 | 1.152 |
| 39 | 0.980 | 0.755 | 0.102 | 0.816 | 0.122 | −0.989 | 1.095 |
| 40 | 0.776 | 0.122 | 0.490 | 0.102 | 0.061 | −1.018 | 1.457 |
| 41 | 0.612 | 0.347 | 0.082 | 0.714 | 0.918 | −1.019 | 0.821 |
| 42 | 0.000 | 0.388 | 0.469 | 0.429 | 0.449 | −1.022 | 1.374 |
| 43 | 0.592 | 0.449 | 0.163 | 1.000 | 0.388 | −1.012 | 1.328 |
| 44 | 0.490 | 0.367 | 0.531 | 0.245 | 0.939 | −1.017 | 0.387 |
| 45 | 0.367 | 0.163 | 0.755 | 0.408 | 0.224 | −1.019 | 1.892 |
| 46 | 0.510 | 0.531 | 0.571 | 0.571 | 0.510 | −1.013 | 1.373 |
| 47 | 0.408 | 0.898 | 0.408 | 0.347 | 1.000 | −1.005 | 0.605 |
| 48 | 0.204 | 0.816 | 0.347 | 0.959 | 0.429 | −1.005 | 1.218 |
| 49 | 0.653 | 0.224 | 0.939 | 0.898 | 0.980 | −1.013 | 1.769 |
| 50 | 0.041 | 0.551 | 0.837 | 0.796 | 0.653 | −1.010 | 1.476 |
| 51 | 0.122 | 0.612 | 0.388 | 0.020 | 0.735 | −1.017 | 0.936 |
| 52 | 0.163 | 0.673 | 0.714 | 0.367 | 0.041 | −1.014 | 1.613 |
| 53 | 0.082 | 0.204 | 0.816 | 0.469 | 0.959 | −1.028 | 1.234 |
| 54 | 0.531 | 0.102 | 0.204 | 0.204 | 0.531 | −1.030 | 0.696 |

## Appendix B

**Table A2.** Pseudo-code of the hybrid PSO-GA.

```
g = 0        * g: generation number
for i = 1 to M do    * M: population (particles) size
        Initialize particles of PSO xᵢ to random values
        xᵢᵇ = xᵢ    * xᵇ: initial information of particle
        Fᵢ = f(xᵢ)    * f: fitness assignment
end for
xᵍᵇ = best{xᵢᵇ; i = 1, … , M}    * xᵍ: initial global best particle
Pop = {x₁, x₂, … , xₘ}
F = {F₁, F₂, … , Fₘ}
<Main Loop>
while do
    {Evaluation Loop 1}
    for i = 1 to M do
    if f(xᵢ) is better than f(xᵢᵇ) then
    xᵢᵇ = xᵢ
    end if
if f(xᵢᵇ) is better than f(xᵍᵇ) then
    xᵍᵇ = xᵢᵇ
    end if
    end for
    {Genetic Operators – Update particles' position}
    Pop ← Selection(Pop, F)
    Pop ← Crossover(Pop, C)      * C: crossover rate
    Pop ← Mutation(Pop, M)    * M: mutation rate
    {Evaluation Loop 2}
    for i = 1 to M do
    Fᵢ = f(xᵢ)
    end for
F = {F₁, F₂, … , Fₘ}
g = g+1
end while
```

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
