# Peer review of "Design and Verification of a Single-Channel Pump Model based on a Hybrid Optimization Technique"

_processes, doi:10.3390/pr7100747_

Round 1

Reviewer 1 Report

The article concerns the curtail problem of the optimization of single blade pump. The presented dates and procedures  are clearly shown.  I suggest to add the sketch of the volute on the figure 15 (it will show the X and Y directions in pump).    It is shame that the radial force and the vibration level was not verified in experimental test, and this is only disadvantage of conducted investigations.  

Author Response

Thank you very much for your comments. I have attached pdf file for answer to reviewer's comment.

Reviewer 2 Report

This study presents a methodology for the optimal geometric design of single-channel pumps for waste-water treatment. The authors use a hybrid particle swarm optimization and genetic algorithm (PSO-GA), coupled with surrogate modeling. The surrogate model is a radial basis neural network (RBNN), which is trained using high-fidelity URANS numerical simulations. The surrogate model is then used to optimize the geometric design variables through particle-swarm and genetic optimization algorithms (PSO-GA). Finally, the authors build a prototype to confirm the hydraulic performance of the optimized model.

Overall, I think the work is interesting, and potentially valuable. However, I have to two major concerns that I believe need to be addressed before the manuscript can be considered for publication:

- The authors need to discuss and clarify what the contribution of this manuscript is, relative their own previous work. Some of the figures (e.g. Figures 1, 2, 5) are the same as, or minor modifications of, some figures from a recent paper by the same authors (Kim et al. Three-objective optimization of a single-channel pump for wastewater treatment, IOP Conf. Series: Earth and Environmental Science, 2019). The author's previous work is mentioned in passing, but the novelties of the present work are not clearly stated.

- The construction of the surrogate model seems to be the main contribution of the paper. And yet there are only superficial details about how this key step is performed. Honestly, it seems like magic to me that the authors can build a black-box surrogate neural network of a complex turbulent flow process, with a 5-dimensional design space, using just 54 realizations. At any rate, the details of how the network is trained, and how it performs as a surrogate model for the full CFD simulation, should be discussed in much more detail.

In my opinion, this work could be published if the authors help potential readers understand their optimization procedure, so that their results can be replicated if so desired. This would imply a major revision of the manuscript to:

Clarify duplicities with respect to the authors' previous work, including new figures. In my opinion, it is not enough to have a simple reference to the original source (among other reasons, because Processes would have to ask for permission to reprint already published figures). Explain with full details how the RBNN surrogate model is built, and how it performs relative to the high-fidelity CFD model. It is not clear to me what the cross-validation error plots in Figure 8 mean, for example. More importantly, I think the authors need to present a clear validation of the surrogate model in the context of the optimization problem. For example, by running a simpler optimization problem where the high-fidelity model can be used directly to evaluate the objective function.

Author Response

(The authors gave the same response as above.)

Reviewer 3 Report

Overall presentation is very well. I have only two suggestions for Authors:

The list of nomenclature is helpful for readers, page 8, 253 line, the lhdesign is the Matlab subrutine and should be clearly described in the mathematical point of view.

Author Response

(The authors gave the same response as above.)

Round 2

Reviewer 2 Report

The revised manuscript is nearly identical to the previous submission. The authors changed Figure 1 and added a few lines describing Matlab functions.

The authors have not addressed my previous comments: there are still figures taken from previous published work, and the construction of the surrogate model and optimization procedure remain black boxes that I find impossible to verify of replicate. Unfortunately, my recommendation cannot be different from that of the first round of reviews.
